# Relationship between Quality of Life and the Complexity of Default Mode Network in Resting State Functional Magnetic Resonance Image in Down Syndrome

**DOI:** 10.3390/ijerph17197127

**Published:** 2020-09-29

**Authors:** Maria Carbó-Carreté, Cristina Cañete-Massé, María D. Figueroa-Jiménez, Maribel Peró-Cebollero, Joan Guàrdia-Olmos

**Affiliations:** 1Institute of Neuroscience, University of Barcelona, 08035 Barcelona, Spain; mpero@ub.edu (M.P.-C.); jguardia@ub.edu (J.G.-O.); 2Serra Húnter Fellow, Department of Cognition, Developmental and Educational Psychology, Faculty of Psychology, University of Barcelona, 08035 Barcelona, Spain; 3Department of Social Psychology & Quantitative Psychology, Faculty of Psychology, University of Barcelona, 08035 Barcelona, Spain; cristinacanete@ub.edu; 4UB Institute of Complex Systems, University of Barcelona, 08007 Barcelona, Spain; 5Los Valles University Center, University of Guadalajara, Jalisco 46600, Mexico; psic.mariafigueroa@hotmail.com

**Keywords:** Default Mode Network, functional connectivity, Down syndrome, resting fMRI, Quality of Life

## Abstract

Background: The study of the Default Mode Network (DMN) has been shown to be sensitive for the recognition of connectivity patterns between the brain areas involved in this network. It has been hypothesized that the connectivity patterns in this network are related to different cognitive states. Purpose: In this study, we explored the relationship that can be estimated between these functional connectivity patterns of the DMN with the Quality-of-Life levels in people with Down syndrome, since no relevant data has been provided for this population. Methods: 22 young people with Down syndrome were evaluated; they were given a large evaluation battery that included the Spanish adaptation of the Personal Outcome Scale (POS). Likewise, fMRI sequences were obtained on a 3T resonator. For each subject, the DMN functional connectivity network was studied by estimating the indicators of complexity networks. The variability obtained in the Down syndrome group was studied by taking into account the Quality-of-Life distribution. Results: There is a negative correlation between the complexity of the connectivity networks and the Quality-of-Life values. Conclusions: The results are interpreted as evidence that, even at rest, connectivity levels are detected as already shown in the community population and that less intense connectivity levels correlate with higher levels of Quality of Life in people with Down syndrome.

## 1. Introduction

The concept of Quality of Life (QoL) that is internationally accepted for people with intellectual disability (ID) was initially presented by Schalock and Verdugo and, until now, has been a reference in the field [1]. This model is characterized by a hierarchical, multidimensional structure and includes both etic (universal) and emic (cultural) components. The eight dimensions of this model have been empirically validated in different cultures and countries [2,3,4] and are organized into three higher-order factors [4]: (1) independence, which includes the dimensions of personal development and self-determination; (2) social Participation, which includes dimensions of interpersonal relations, social inclusion, and rights; and (3) well-being, which includes the dimensions of emotional well-being, physical well-being, and material well-being. This concept has become a social construct that provides a reference for assessing the impact of processes and services addressed to this population [5,6,7,8], and, together with the studies related to health levels [9,10], they have become one of the most mentioned topics in this field. Accordingly, most of works focused on QoL analyze the characteristics of the inputs and in the measurement procedure in itself; however, little is investigated about other effects emerged related with QoL. This is the main question that we pursued in this work. Our main purpose was to examine if there is some effect in brain connectivity that could be related to the QoL dimensions. 

To explore the brain functioning, the neuroimaging techniques are required, such as functional Magnetic Resonance Imaging (fMRI). The fMRI measurements provide information about functional connectivity, defined as a temporal correlation between spatially remote brain regions [11]. Brain connectivity between distant regions can also be examined when an individual is in a resting state (rs-fMRI), detecting the spontaneous low frequencies (<0.01 Hz) through the Blood Oxygen Level Dependent (BOLD) signals [12,13]. In rs-fMRI, the person is not concentrate to external stimulus, and a specific network of brain regions is activated, called Default Mode Network (DMN) [14]. The DMN was identified by using block-design positron emission tomography (PET) studies when in passive tasks the activity increased in comparison to active tasks [15]. There are different patterns where DMN is detected, but we are focused in the high intrinsic activity identified when the subject is doing nothing (without any task during scanner), with the eyes closed or looked at a fixed point (e.g., see References [16,17]). 

There is evidence of functional and structural differences on people with ID, specifically with Down syndrome (DS); the lowest brain volume and the lowest activity (activation) recorded by fMRI seems to be typical of this population [18]. Based on previous works focused on DS, the connectivity of the DMN to other brain regions shows different profiles in contrast to healthy control groups. The main characteristics of the connectivity in DS are the increased positive networks identified (i.e., hyper-connectivity), even in regions that, in control groups, are negative, and the decreasing of anti-correlation [19,20,21]. These data can explain the altered brain functioning of DS, due to the fact that anti-correlation is necessary to organize and segregate networks in the healthy brain [22].

Related to the study of brain connectivity, there is a growing interest based on the knowledge of the human brain from the perspective of a complex network [23,24]. The study of the complex network implies the mathematical analysis known as graph theory, which defines the brain architecture connectivity by using quantitative data. The graph theory is represented by a group of vertices, called nodes, connected between them by links, called edges (Figure 1). In fMRI, nodes are anatomical defined and known as a regions of interest (ROIs) on the basis of specific atlases (e.g., automated anatomical labeling (AAL) template and Montreal neurological institute (MNI)). The edges represent the dependence between nodes and can be differentiated based on their directionality and weight. Thus, based on graph theory, the functional connectivity depends on the magnitude of the correlation between the time-series of two regions (ROIs) of the brain [25,26], computed by a set of indicators defined in Reference [27] (Figure 1). 

In the current paper, we want to introduce this approach due to previous works [19] provide a comprehensive information related to the segregation of the brain in DS, as well as the remarkable data obtained through graph theory in papers focused on other populations groups, for example, in Alzheimer’s Disease [28]. In the study of Anderson et al. [19], they analyzed the clustering coefficient (functional segregation measure) and obtained higher scores in DS than control group, showing a weak local organization of brain networks. The work with the Alzheimer’s disease population obtained remarkable results in relation to the small-world measure. The small-world organization is defined by the optimal balance between clustering coefficient and the characteristic path length (functional integration measure) [29]. In Alzheimer’s disease study, the characteristic path length was low, similar to the control group, and no significant differences were observed. Nevertheless, the clustering coefficient were lower in Alzheimer’s disease than in the control group, mainly in the left and right hippocampus. These results are interpreted as a loss of small-world and, in consequence, altered brain connectivity [28].

Concurrently these studies about connectivity and DS, concepts related to QoL, have been examined by using fMRI techniques, mainly in resting state (rs-fMRI). Focusing on QoL, the works of Luo and their colleagues [30,31] about happiness are a clear evidence of the contributions of fMRI techniques. In these studies, the authors have shown that unhappy people present alterations in the local synchronization of intrinsic brain activities in some areas (prefrontal cortex, temporal lobe, limbic system, and subcortical regions). Moreover, in comparison with happy people, there is an increasing in the functional connectivity is some areas of the DMN associated with an increase of ruminate. Similar work is presented by Kraft et al. [32] about the relationship between four QoL domains (physiological, psychological, social, and environmental) and the functional connectivity of the DMN of healthy working females. 

There are few works that use the analysis of complex networks to study the relation between some disease and QoL. Nevertheless, the initial studies that focus on rs-fMRI and graph theory to investigate QoL encourage us to use these techniques in people with ID. For example, in children with focal epilepsy [33], the results showed a relation between functional segregation, measured by clustering coefficient and modularity, and social functioning, that implies multiple higher-order brain functions. Nevertheless, they did not find an association between global graph indices and well-being. The authors suggest that the measuring instrument was not specific enough, and it is required for these kinds of studies. 

Thus, the objective of this paper is to analyze a set of indicators of complex networks defined in Rubinov and Sporns [27], to study if QoL presents some relation with the organization and characteristics of brain networks, analyzed by functional integration and segregation measures. The first hypothesis in the present work is that the rs-fMRI brain network in DS people shows networks more complex in comparison with the same structure estimated in the control group; the second hypothesis is that there is a negative relation between QoL scores and brain complexity, so the higher values of QoL are related to a less-complex brain connectivity network and vice versa. We expect to contribute to the knowledge about brain functioning of DS population and to provide initial data about QoL and connectivity. The use of graph theory can provide a widely information about the architecture of the brain of people with DS in relation their QoL, so we will be enhancing one of the most relevant constructs in the ID field.

## 2. Methods

### 2.1. Participants

The sample was composed of 22 persons with DS (M = 25.5; SD = 5.17), and 22.7% were women. It was a convenience sample, but we followed the recommendation of Friston [34], i.e., about 16 subjects in studies of fMRI. The recruitment was performed through several services that attend people with intellectual disabilities in Spain (63.6%) and Mexico (36.4%). Regarding the inclusion criteria, the age had to be between 16 and 35 years old, and all of them had to have a formal diagnosis of DS. The exclusion criteria were evidences of other comorbid diagnoses implying cognitive dysfunction or impossibility of obtaining consent from legal tutors. Finally, the presence of medication affecting cognitive functions was also an exclusion criterion. Due to these criteria and excessive movements into scanner, thirteen participants of the initial sample (*n* = 35) were excluded. In reference to the intellectual-disability level of the final sample, most of them had the diagnosis of moderate (40.9%), and the rest were mild (36.4) or borderline (4.5%). 

A control group (*n* = 22) was included to compare the indicators of complex networks analyzed in the DS population. These subjects were obtained from the Human Connectome Project (http://www.humanconnectomeproject.org/), specifically from the open-access dataset Autism Brain Imaging Data Exchange I (ABIDE I). The ABIDE I is an image repository comprising 17 international sites; it collects structural and rest fMRI scans from people with autism spectrum disorder and healthy control groups. All data, including the phenotypic datasets and the protocol of acquisition parameters, are available at http://fcon_1000.projects.nitrc.org/indi/abide/abide_I. Only the control group of the ABIDE I dataset was used, and the subjects were selected to be matched with DS sample by chronological age (M = 24,68; SD = 4.90; maximum 2 year of difference in some subject) and gender (22.7% were women). The control subjects were from the following universities: University of Utah School of Medicine (*n* = 15), University of Michigan-Sample 1 (*n* = 2), University of Michigan-Sample 2 (*n* = 2), and the Carnegie Mellon University (*n* = 3). No statistical differences were found in relation to age (t = 0.568; df = 42; *p* = 0.573). 

### 2.2. Instruments

These data belong to a greater amplitude protocol than the one presented here. The part analyzed here was characterized by a first questionnaire of usual sociodemographic data (age and gender), to subsequently administer the following instruments to the sample of DS people.

#### 2.2.1. Checklist for MRI-Scanner

To ensure maximum safety for participants, a doctor’s approval was requested. The authorization and doctor’s consent include a review to avoid the presence of elements not suitable for resonance (metal prostheses, electronic devices, dentures or other types of implants, skin tattoos, etc.).

#### 2.2.2. Quality-of-Life Assessment

Based on the objectives of this study, the QoL was measured only in the DS group. This assessment was made by using the Spanish version of the Personal Outcomes Scale (POS) [35]. This scale comprises the eight dimensions and the three factors of Schalock and Verdugo QoL model [1]. The Spanish version of the POS includes self-report and report of the others (professional version or family version). In all situations, the content of the items is the same; the unique difference is the wording. Nevertheless, for this study, we only administered the report of the others; when it was possible, we interviewed the families, but, if not, we administered the POS to professionals of the services. The reliability Spanish POS adaptation [35] provides appropriate values for the first-order domains and, particularly, for the second-order factors, with α values higher than 0.82. Moreover, the construct validity analysis provides an adjustment of the theoretical model with regard to the three sources of information, particularly regarding the professionals’ assessments. 

### 2.3. Procedure

The applied protocol was approved by the Bioethical Committee of the Universitat de Barcelona (16/03/2017). Informed consent of the tutors in legal charge of every person with DS was obtained. Moreover, informed consent was obtained from the participants with DS. 

All the participants were evaluated in two register sessions. The first was dedicated to the questionnaires administration and the image acquisition, and the second session was dedicated to another part of the general research design. The administration sequence was the same for all the participants, and the scales referenced above were administered first, to avoid fatigue bias.

#### 2.3.1. Image Acquisition 

The MRI-scanner in Mexico and Spain were very similar, in order to avoid differences in the recording. Two Philips Ingenia 3.T system models were used (one located at the Laboratorio Clínico, Centro Integral de Diagnóstico Médico of Guadalajara’s Grupo Río Center in Jalisco and the other at the Fundació Pasqual Maragall in Barcelona). All the participants performed an fMRI recording sequence: T1, T2, Flair, and 6-minutes resting-state. The instructions for the participants were the following: Try to stay still and without movements, remain awake, and keep your eyes open and fixed on the cross symbol on the screen. During all the recording, except the 6 minutes of resting, participants could choose music to hear. A T1-weighted turbo field echo (TFE) structural image was obtained for each subject with a 3-dimensional protocol (repetition time (TR) = 2300 ms, echo time (TE) = 2980 ms, 240 slices, and field of view (FOV) = 240 × 240 × 170). The image acquisition was in the sagittal plane. For the functional images, a T2 *-weighted (BOLD) image was obtained (TR = 2000 ms, TE = 30 ms, FOV = 230 × 230 × 160, and voxel size = 3 × 3 × 3 mm, 29 slices). The image acquisition was in the transverse plane.

Regarding the control group, the acquisition was performed in different institutions of the United States. As in the case of the DS group, all the participants performed fMRI recording sequence: T1, T2, Flair, and between 6 and 9 minutes resting-state. The repetition time (TR) in all cases was 2000 ms, and the voxel size was different for every protocol. Moreover, due to the extra minutes in resting, in all the cases in the control group, the number of volumes was greater than in the DS group (oscillating between 240 volumes and 300). Therefore, we used only the first 220 volumes corresponding to the ones used in the DS group.

#### 2.3.2. Image Preprocessing and ROIs Extraction

The structural and functional image data were analyzed by using a FSL (FMRIB Software Library v5.0), with preprocessing pipeline adapted under authorization from Diez et al. [36]. However, we adjusted its parameters, to fit our experimental data. We included motion correction for this special population. T1 images were reoriented, to match the same axes as the templates, and a resampled AC–PC (Anterior Commissure-Posterior Commissure) aligned image with six degrees of freedom (df) was created. All non-brain tissue was removed, to obtain an anatomic brain mask that would be used to parcel and segment the T1 data images. The final step involved registering our structural data images to the normalized space, using the Montreal Neurological Institute reference brain based on the Talairach and Tournoux coordinate system [37].

In relation to fMRI data, the first 10 volumes were discarded for correction of the magnetic saturation effect, and the remaining volumes were slice-time corrected for temporal alignment. All voxels were spatially smoothed with a 6 mm FWHM (Full Width at Half Maximum) isotropic Gaussian kernel, and, after intensity normalization, a band pass filter was applied between 0.01 and 0.08 Hz [38], followed by the removal of linear and quadratic trends. Finally, the functional data were spatially normalized to the MNI152 brain template. 

The automated anatomical labeling (AAL) atlas [39] was used to define the regions of interest (ROIs). This atlas contains 45 cortical and subcortical areas in each hemisphere (90 areas in total), which are alternatively interspersed (available by request). 

To acquire the full signal of a given ROI, a computation of the average over the entire time-series of all the voxels of a given brain area following the AAL atlas was made. In this study, only the DMN region was included, and therefore only the usual 24 ROIs that define DMN were identified. Table 1 shows the number of ROIs used in the study, their correspondence with the AAL atlas, and the region name, following the classification proposed by Huang et al. [40]. As we can see, the 24 ROIs added in the study can be divided in three regions, namely DMN, DMN anterior, and DMN ventral, in accordance with Yeo et al. [41].

However, the analyses in this study are made free of the subnetworks that conform the DMN. In addition, Figure 2 represents the exact localization of each analyzed ROI.

### 2.4. Statistical Analysis

To extract the indicators of network’s complexity, we used a specific script base on the following R libraries: readxl; NetworkToolbox; QuACN; corrplot, MASS, clusterGeneration, qgraph, GeneNet, corpcor, longitudinal, fdrtool, circlize, and clValid, factoextra. Table 2 shows the selection of the most interesting indicators [29] estimated in both groups.

Subsequently, the medians values of each group were compared, to analyze the first objective and identify if our expectation was confirmed. For this, the non-parametric Mann–Whitney test was used to avoid the effect of the small sample size. In addition, the adapted Levene test for two groups was used to establish whether, according to our hypotheses, the DS group of people is characterized by a greater variability in the distributions of the indicators of complexity, as compared to the control group. Finally, to determine if in the case of the DS group there is a negative relationship between the QoL values and the distributions of the complexity indicators, the estimation of a stepwise regression model was used, using the QoL values as an exogenous variable and the indicators of complexity as exogenous variables, analyzing the best model by adjusting the highest value of *R*^2^ and the lowest value of Akaike Information Criteria (AIC). All analyses were performed by using R libraries.

## 3. Results

First, Table 3 shows the descriptive statistics of the complexity indicators for each group, including the significance values of the one-sided contrast (DS > Control) that were estimated by using the non-parametric Mann–Whitney test, to avoid the effect of the reduced sample size.

According to the previous results, the structure of the complexity networks seem to present more complexity (DS > Control) according to the observed distribution of the Global Clustering Coefficient, Standard Deviation of the path length, and Dunn Index. These results are congruent with the conception of major complexity structure in DS group. In addition, for the path length and small-worldness, the results are congruent with the other direction (Control > DS). This result is connected with the idea of a more stable network for the control group. No other indicators showed statistically significant differences.

Continuing with the first objective of this work, it was analyzed if the variances of both groups were equal, since it was expected to find greater variability in the DS group. For this, the Levene test, adapted for two groups, was used. The results indicated that, in most of the indicators, the variance was higher in the DS group, especially significant results in Global Clustering Coefficient, modularity, path length, mean path length, and SD path length. Consequently, we can assume a greater variability of the distributions in the DS group.

In relation to the second objective, to assess the possibility of attributing this greater variability of the DS group to the distribution of the QoL, Pearson’s correlation values were estimated from QoL dimensions and graph theory measures (Table 4). To facilitate the interpretation of the correlations, the data are graphically presented in Figure 3. We are aware that the small sample makes it difficult to estimate parameters of the linear regression model, but we have included them in order to show the statistically significant impact of those indicators of brain complexity networks in relation to QoL. As a consequence, Table 5 shows the parameters values of the best linear model adjusted by *R*^2^ and the AIC that predicts each of the eight dimensions described to assess QoL.

Only one dimension is not available to adjust a liner model (i.e., rights). The other seven dimensions can be predicted by using some complexity indicators as exogenous variable in a linear model. The *β* parameter estimations results have some of them as negative parameters. This result is congruent with the negative relationship between complexity and QoL. The positive values of β are related to indicators of stability of the brain connectivity network. In DS people, the stability of the network is connected with high values of QoL. 

## 4. Discussion

The data analyzed indicate that, briefly, the complexity indicators values of the connectivity networks in persons with DS tend to show less stable and more complex networks than the ones characteristic of the control group. Moreover, more variability is found in these indicators in the case of the DS group, a finding which is congruent with the usual conception of less inter-individual variability in these persons. Finally, it was possible to adjust some lineal models in order to predict the eight dimensions of the QoL model of Schalock and Verdugo [1] from some complexity indicators aforementioned. 

The fact that there are no previous works in the field of DS persons and brain connectivity networks complicates the establishment of a discussion in the literal sense of the word. However, some details exist that, in our opinion, should be highlighted. 

In first place, the conception that the connectivity patterns study with fMRI signal in resting situation seem to be an adequate mechanism for the study of the brain functioning in people who have cognitive difficulties and which a task realization could compromise the result. Much has been discussed [9] on this issue, and, in the case of people with DS, it seems that the idea should be maintained. The resting situation in a brain signal recording is a good cognitive evaluation option. Obviously, we must know in more detail the behavior of many other indicators of the complexity brain network, but those used in this work provide relevant information on the matter.

Secondly, the complexity differences found between both groups (DS vs. control) are extraordinarily suggestive for the classification of intellectual disability. The more complex and less stable networks in DS are than in control persons indicates a differential functioning to consider in future investigations. The existence of higher values of Global Clustering Coefficient, the SD of path length, complexity, and Dunn’s index would indicate this. The sample of DS is characterized, based on these indicators, by a more complex network in its connections between ROIs. However, the values of characteristic path length or small-worldness would not confirm this conception, since the statistical significance obtained shows an effect contrary to that expected; moreover, the rest of the indicators, which have not shown statistically significant differences between the two groups, would not support our expectations either.

A possible explanation for this paradoxical effect can be found in the variability values of the observed distributions (for example, SD path length), which shows enormous variability in the DS sample, compared to control group. This issue makes it difficult to establish a clear pattern. Despite this, our results point, albeit no sharply, to the expected effect in the pattern of the network in DS is more complex than in the control group.

Although the correlations obtained do not allow to identify easily any behavior pattern between complexity and QoL, they do identify some values associated with a strong relationship between both concepts. For example, the correlation between triangles and Personal Development factor (−0.558), among others. This implies that there is evidence of a relationship between both constructs, the QoL factors and the behavior of a functional connectivity network in resting. Obviously, this relationship is neither massive nor regular in all cases, but it suggests that high values in the QoL factors are associated with a lower complexity structure in the network of functional connectivity. Unfortunately, this question cannot be verified in the control group, since we do not have these evaluations. However, it would be expected that, in the case of the controls, the relationship would be much less intense and even null, since the network in the control group is less segregated and more homogeneous. If we try to identify which complexity values could be predictors of the distributions of QoL dimensions, the question presents a clear limitation regarding the sample size. However, for descriptive purposes, the values and signs of the standardized regression coefficients indicate the same behavior. We could think, in the future, of using the values of these indicators as a predictor of QoL, although our results are not consistent with a previous work [19]. In general, both works point to the idea of the relationship between complexity structure, but we identify more evidence of relationship than Nawani et al. [33]. The discrepancy can be explained by the different number of ROIs studied and the study design. Moreover, the comparison between both works is difficult, since their samples are children and the psychometric scales are too specific. The inconsistency between their results and ours can be explained by more methodological than conceptual elements.

The lineal models in Table 4 show that it will be feasible to think in a possible prediction of the QoL values from the complexity data. This option is not incompatible with the more psychological conception of the QoL measure, because, although the values of *R*^2^ are very important, the models do not explain totally the values variability observed in QoL. Therefore, it is true that the complexity indicators are related (inversely) with QoL and that the stability indicators are also related (positively); however, we still have a big margin of unexplained variance, so this is not the only system that could be proposed. Neither the intragroup variability (in the DS case) nor the intra-individual variability allow us to think that it is. 

A point derived from that mentioned above is that there are a lot of pathologies that are not directly linked with cognitive impairment and not related with intellectual disability. It will be very useful to identify if these types of patterns can be constant throughout other diagnoses, and in the case that they are not stable, establish the possible differences. There has been a lot of discussion about the possibility of a cognitive functioning biomarker. In our opinion, the study of connectivity patterns can be an answer to this question. At a minimum, the comparison between DS and control groups and our data indicate this. In our opinion, we are taking the first steps in this type of approach, and we are not in a position to draw very strong conclusions, not only because of the sample size, but also because we have selected only a few very known indicators that do not exhaust all the complex systems study possibilities.

Finally, in this work, there are various limitations that we can group into three aspects. The first is about methodological deficits; basically, the lack of larger sample sizes and the availability of equivalent QoL measures in the control group. Second, there is a more conceptual limitation since we only used a part of the existing complexity indicators, so that the study possibilities are much greater than those presented here. It is true that we used the most common ones, but this does not exclude the need for much more exhaustive analysis. Third, we do not have information about sociodemographic data of the control group, and we have not been able to establish comparisons on these variables between the two groups.

## 5. Conclusions

From the above considerations, we can briefly highlight the following conclusions. The functional brain connectivity networks estimated by recording in rs-fMRI show different patterns of complexity and stability between DS and control people. The differences focus on the fact that the networks in DS are more complex and less stable than in control people. Furthermore, the indicators of the connectivity patterns and stability networks show, in general, greater variability in the DS group, as compared to the control group. This should be interpreted as the existence of inter-individual variability in DS much greater than in the case of controls. Finally, the patterns of the functional connectivity network can be used to predict the QoL dimensions values, with the exception of the rights dimension, which suggests that, at higher levels of complexity, less value is obtained in the QoL dimensions. On the other side, the greater the stability of the network is, the greater the values in the dimensions of QoL are.

In the near future, we will have to study other complexity indicators and stability, to confirm this issue, as well as analyze whether there are neurofunctional properties of the ROIs studied to assess this type of pattern in a more functional way.

## Figures and Tables

**Figure 1 ijerph-17-07127-f001:**
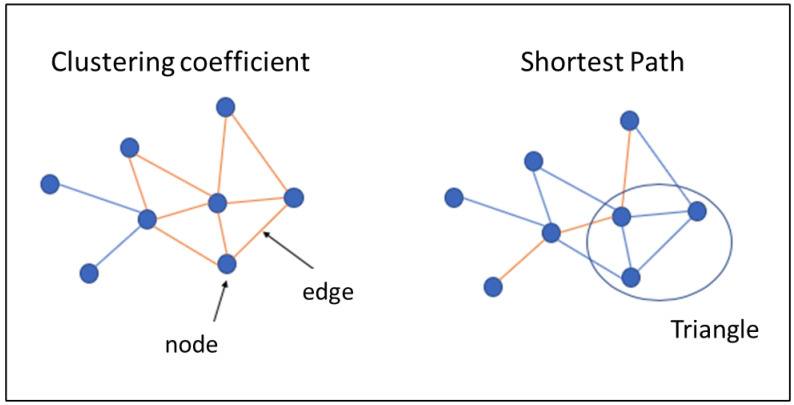
Example of indicators of complex networks.

**Figure 2 ijerph-17-07127-f002:**
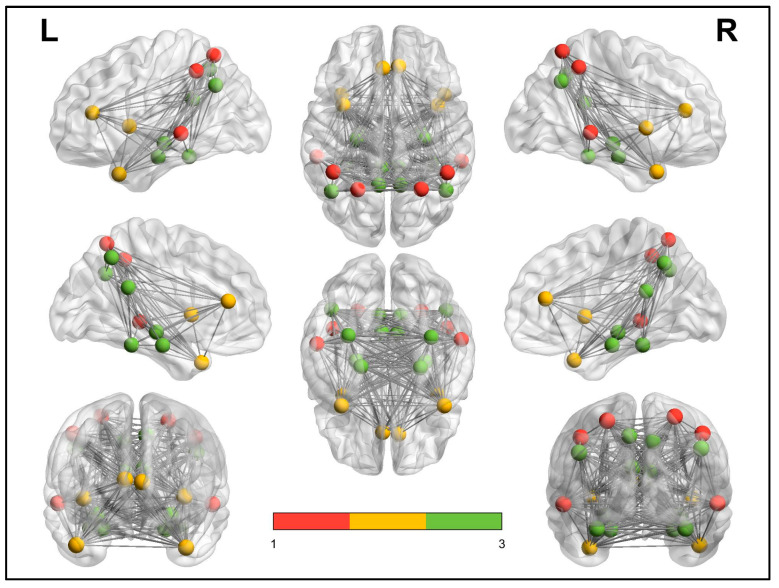
Exact localization of each ROI described in the Table 1. Red: DMN. Yellow: DMN anterior. Green: DMN ventral. The edges are simulated to facilitate the representation.

**Figure 3 ijerph-17-07127-f003:**
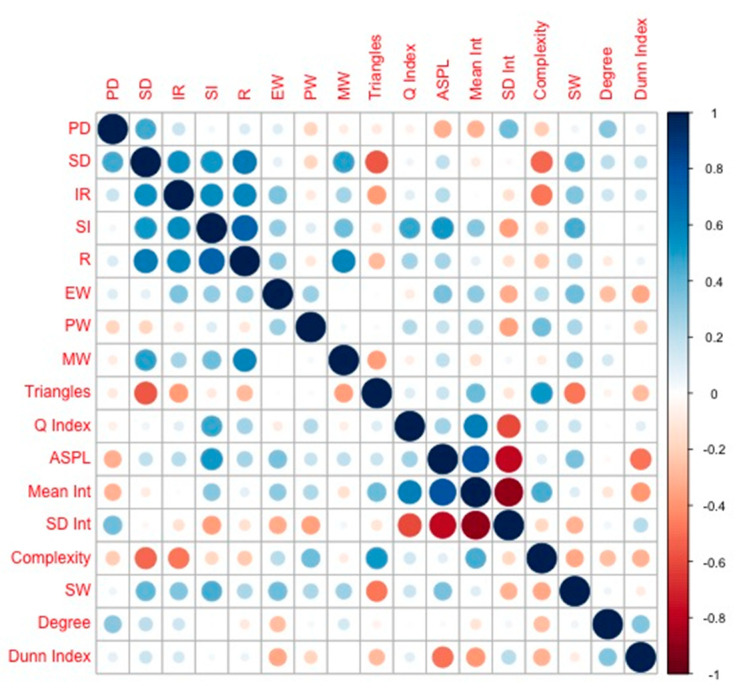
Corrplot of correlation matrix between Quality-of-Life (QoL) dimensions and graph theory measures. PD = personal development, SD = self-determination, IR = interpersonal relations, SI = social inclusion, EW = emotional well-being, PW = personal well-being, MW = material well-being, ASPL = characteristic path length, Mean Int. = Mean Integration, SD Int. = Standard Deviation Integration, and SW = small-worldness.

**Table 1 ijerph-17-07127-t001:** Relationship of regions of interest (ROIs) for the construction of the Default Mode Network (DMN) and subnetworks considered according to the AAL90 atlas.

DMN		DMN Anterior		DMN Ventral	
ROI	Region Name	ROI	Region Name	ROI	Region Name
59	Parietal_Sup_L	29	Insula_L	35	Cingulum_Post_L
60	Parietal_Sup_R	30	Insula_R	36	Cingulum_Post_R
61	Parietal_Inf_L	31	Cingulum_Ant_L	37	Hippocampus_L
62	Parietal_Inf_R	32	Cingulum_Ant_R	38	Hippocampus_R
85	Temporal_Mid_L	87	Temporal_Pole_Mid_L	39	ParaHippocampal_L
86	Temporal_Mid_R	88	Temporal_Pole_Mid_R	40	ParaHippocampal_R
				55	Fusiform_L
				56	Fusiform_R
				65	Angular_L
				66	Angular_R
				67	Precuneus_L
				68	Precuneus_R

**Table 2 ijerph-17-07127-t002:** Complex network indicators used in the study.

Indicators	Description	Calculations
Degree	Number of links connected to all nodes. Important marker of network development and resilience.	$$k_{i}=\sum_{j\in N}a_{ij}$$ *N* is the set of all nodes in the network, (*i,j*) is a link between nodes *i* and *j* (*i,j*∈*N*). *a_ij_* is the connection status between i and *j*: *a_ij_* = 1 when link (*i, j*) exists (when *i* and *j* are neighbors); *a_ij_* = 0 otherwise (*a_ii_* = 0 for all *i*).
Number of Triangles	A basis for measuring segregation. Number of triangles around a node *i.*	$$t_{i}=\frac{1}{2}\sum_{j,h\in N}a_{ij}a_{ih}a_{jh}$$
Global Clustering Coefficient	Prevalence of clustered connectivity around individual nodes.	$$C=\frac{1}{n}\sum_{i\in N}C_{i}=\frac{1}{n}\sum_{i\in N}\frac{2t_{i}}{k_{i}\left(k_{i}-1\right)}$$ where *C_i_* is the clustering coefficient of node *I* (*C_i_* = 0 for *k_i_* < 2).
Characteristic Path Length	The characteristic path length is a global measure of the network, i.e., there is only one value for the entire network. It consists of the average path length of each node in the network.	$$Li\;=\;\sum_{i\epsilon N}\left(\frac{1}{n-1}.\;\underset{j\epsilon N,j\neq i}{\sum }dij\right)$$ where *n* is the number of nodes involved and *d_ij_* is the shortest path length between node *i* and *j*.
Modularity	Where the network is fully subdivided into a set of non-overlapping modules M, and *e_uv_* is the proportion of all links that connect nodes in module *u* with nodes in module *v*.	$$Q=\sum_{U\in M}\left[e_{uu}-{\left(\sum_{v\in M}e_{uv}\right)}^{2}\right]$$
Small-Worldness	Measures an optimal balance of functional integration and segregation on the networks.	$$S=\frac{\frac{C}{C_{rand}}}{\frac{L}{L_{rand}}}$$ where *C* and *C_rand_* are the clustering coefficients, and *L* and *L_rand_* are the characteristic path lengths of the respective tested network and a random network.

Complexity: the number of nodes and alternative paths that exist within a specific network. Mean path length: the path length of a node *i* (*Li*) is the average number of edges that must be crossed to go from node *i* to the rest of nodes in the network. Standard deviation of the path lengths: variability measure of the path length of each node in the network. Dunn Index: measure of sets of clusters that are compact, with a small variance between members of cluster. The higher the Dunn index value, the better the clustering is.

**Table 3 ijerph-17-07127-t003:** Statistical descriptive indexes for DS and control groups.

Indicators	Group	*n*	Mean	Mean Rank	SD	Significance
Global Clustering Coefficient	Down *	10	0.9349	21.80	1.39454	0.015
	Control	22	0.4579	14.09	0.04490	-
Number of Triangles	Down	22	858.68	20.77	701.804	0.186
	Control	22	1049.50	24.23	837.193	-
Modularity	Down	22	0.3792	21.41	0.27243	0.286
	Control	22	0.4467	23.59	0.03244	-
Characteristic Path Length	Down	22	1.1342	14.73	6.32448	<0.001
	Control	22	3.8778	30.27	0.48123	-
Mean Path Length	Down	22	0.5830	20.18	0.15949	0.115
	Control	22	0.6471	24.82	0.06442	-
SD Path Length	Down	22	0.2909	26.73	0.17160	0.014
	Control	22	0.1511	18.27	0.02040	-
Complexity	Down	22	0.5867	27.14	0.05813	0.008
	Control	22	0.5272	17.86	0.10270	-
Small-Worldness	Down	22	1.6735	19.23	0.12892	0.045
	Control	22	1.7235	25.77	0.11505	-
Degree	Down	22	4.1976	23.23	0.17693	0.353
	Control	22	4.1490	21.77	0.14788	-
Dunn Index	Down	22	0.4945	32.50	0.10255	<0.001
	Control	22	0.1811	12.50	0.22281	-

* Estimation of Global Clustering Coefficient no converge.

**Table 4 ijerph-17-07127-t004:** Correlation matrix between QoL dimensions and graph theory measures.

	PD	SD	IR	SI	R	EW	PW	MW	Trian-Gles	Q Index	ASPL	Mean Int.	SD Int.	Com-Plexity	SW	Degree
PD	-	-	-	-	-	-	-	-	-	-	-	-	-	-	-	-
SD	0.454 *	-	-	-	-	-	-	-	-	-	-	-	-	-	-	-
IR	0.168	0.543 **	-	-	-	-	-	-	-	-	-	-	-	-	-	-
SI	0.041	0.515 *	0.554 **	-	-	-	-	-	-	-	-	-	-	-	-	-
R	0.116	0.621 **	0.577 **	0.722 **	-	-	-	-	-	-	-	-	-	-	-	-
EW	0.101	0.089	0.348	0.293	0.307	-	-	-	-	-	-	-	-	-	-	-
PW	−0.188	−0.185	−0.091	0.106	−0.109	0.276	-	-	-	-	-	-	-	-	-	-
MW	−0.080	0.483 *	0.251	0.370	0.586 **	−0.005	0.038	-	-	-	-	-	-	-	-	-
Triangles	−0.093	−0.558 **	−0.372	−0.090	−0.273	−0.014	0.029	−0.364	-	-	-	-	-	-	-	-
Q Index	−0.069	0.052	0.094	0.460 *	0.266	−0.088	0.222	−0.075	0.106	-	-	-	-	-	-	-
ASPL	−0.320	0.196	0.219	0.519 *	0.260	0.357	0.174	0.192	0.165	0.265	-	-	-	-	-	-
Mean Int.	−0.305	−0.087	−0.017	0.322	0.085	0.302	0.233	−0.136	0.385	0.609 **	0.785 **	-	-	-	-	-
SD Int.	0.378	−0.032	−0.135	−0.363	−0.131	−0.322	−0.359	0.042	−0.116	−0.598 **	−0.773 **	−0.882 **	-	-	-	-
Complexity	−0.212	−0.511 *	−0.475 *	−0.177	−0.223	0.213	0.375	−0.080	0.511 *	0.148	0.094	0.449 *	−0.176	-	-	-
Entropy	0.132	0.293	0.053	−0.177	−0.109	−0.274	−0.083	0.140	−0.291	−0.401	−0.047	−0.323	0.198	−0.454 *	-	-
SW	0.055	0.409	0.336	0.442 *	0.242	0.373	0.240	0.272	−0.473 *	0.166	0.352	0.100	−0.301	−0.330	-	-
Degree	0.324	0.204	0.152	0.007	−0.091	−0.264	0-037	0.131	−0.055	0.018	−0.030	−0.118	0.047	−0.269	0.051	-
Dunn Index	0.089	0.165	0.131	0.039	0.061	−0.333	−0.181	−0.006	−0.275	0.095	−0.483 *	−0.385	0.218	−0.309	−0.084	0.334

PD = personal development, SD = self-determination, IR = interpersonal relations, SI = social inclusion, EW = emotional well-being, PW = personal well-being, MW= material well-being, ASPL = characteristic path length, Mean Int. = Mean Integration, SD Int. = Standard Deviation Integration, and SW = small-worldness. * *p* < 0.05; ** *p* < 0.001.

**Table 5 ijerph-17-07127-t005:** Parameter estimation (*β*) of best stepwise linear models for each QoL dimension. All cases estimated with *p* < 0.01.

Complexity Indicators	Personal Development	Self Determination	Interpersonal Relations	Social Inclusion	Rights	EmotionalWell-Being	PhysicalWell-Being	MaterialWell-Being
	*R*^2^ = 0.630 *AIC* = 19.93	*R*^2^ = 0.641 *AIC* = 21.49	*R*^2^ = 0.251 *AIC* = 24.13	*R*^2^ = 0.820 *AIC* = 35.61		*R*^2^ = 0.391 *AIC* = −4.26	*R*^2^ = 0.443 *AIC* = 16.95	*R*^2^ = 0.317 *AIC* = 42.38
Global Clustering Coefficient	-	-	-	-	-	-	-	-
Number of Triangles	0.002	-	-	-	-	-	-	-
Modularity	-	-	-	10.122	-	−1.734	-	-
Characteristic Path Length	−0.569	-	-	0.522	-	-	-	0.396
Mean Path Length	51.340	18.665	-	-	-	3.289	−12.852	−14.821
SD Path Length	22.493	-	-	14.314	-	-	−12.605	-
Complexity	−75.633	−42.367	−18.081	−49.936	-	-	19.136	-
Small-Worldness	-	-	-	-	-	-	-	-
Degree	-	-	-	−6.374	-	-	-	-
Dunn Index	-	-	-	-	-	-	-	-

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
