# Peer review of "Relationship between Quality of Life and the Complexity of Default Mode Network in Resting State Functional Magnetic Resonance Image in Down Syndrome"

_ijerph, 2020, doi:10.3390/ijerph17197127_

Round 1

Reviewer 1 Report

Thank you for the opportunity to review your manuscript, entitled "Study of the relationship between Quality of Life and the complexity of Default Mode Network in Resting State Functional Magnetic Resonance Image in Down Syndrome." Understand the topological architecture of the brain, particularly in populations such as ID and DS, may provide key insights into effective interventions. To this end, manuscripts such as this one are welcome and necessary contributions to the extant literature base. 

However, for me, this present work falls short, though I believe that it can be reasonably edited with minimal effort. The two major concerns I have (detailed further below) are the rationale for the approach and and the description of the methodology. In the broadest sense, it is not clear why you are choosing graph theory (versus any other kind of rs-fMRI analytic approach). Outlined below are my concerns:

Introduction:

My major concern with the introduction is that it does not lead to the natural conclusion that this study was necessary. Much of the introduction is not specific to DS, nor are rs-fMRI/graph theory findings from DS populations discussed. There is limited connection to network topology and QoL. And, most curiously, there is limited identification of the major gaps in our understand of DS when using rs-fMRI. This is especially curious given your own recent systematic review and the conclusions from that review on this exact topic.

As a reader, the organization of the introduction is as follows:

  1. The social construction of QoL.
  2. Services for ID may impact QoL and brain function.
  3. General description of rs-fMRI with a brief mention that ID/DS populations have low volume and activity.
  4. QoL (and potentially related topics, e.g. happiness) and fMRI.
  5. Introduction to graph theory
  6. Graph theory in AD/schizophrenia (non-specific findings).
  7. Purpose: Based on all of these disconnected pieces, we want to study graph theory  in DS.

Hopefully, you can see that this overarching structure to the introduction does not lead to the natural conclusion that this study is necessary. It would be more helpful to have an introduction organized in the following way:

  1. The social construction of QoL and specific evidence that QoL is decreased for individuals with DS.
  2. A brief introduction to rs-fMRI as a means to understand how the brain functions and a summarised findings from DS populations.
  3. A brief introduction to graph theory as a means to understand how the brain is organized and summarised findings from ID populations on graph theory measures. This should briefly include an interpretation of what certain aspects mean (i.e., Studies x,y, and z found decreased small-worldedness in <some population> compared to controls. Decreased SW suggests more diffuse network connectivity, in contrast to more modular connectivity in control populations meaning <your interpretation of decreased SW>). In the absence of, in your estimation, relevant literature from similar populations, then using AD/schizophrenia is appropriate but should still provide some measure of interpretation.
  4. An identification of the critical gaps in our understanding. Namely, here, the lack of information on the relationship between QoL and network topology in DS populations. Please note: As currently written, your rationale indicates that you believe QoL is causative of functional connectivity patterns in DS populations and that network complexity impacts functional connectivity. Neither of these are likely accurate statements. QoL may be related to but not causative of brain connectivity. Likewise, network complexity is a representation of the functional connectivity but does not impact it (as evidenced by the fact that ROI-to-ROI FC must be computed first).
  5. An identification of how the present study seeks to fill one or more these critical gaps and what the a priori hypotheses are.

Methods:

I have several concerns with how the methods are described. These should be easily remedied.

  1. You only describe the "structural imaging" pre-processing. There is no description of the methods taken to pre-preprocess the functional data. Given the myriad of choices that need to be made when processing rs-fMRI data (slice time correction, susceptibility distortion correction, bias correction, motion correction, filtering, etc), these need to be clearly described and appropriately cited for each step. Especially because you adapted the procedure from Diez et al., you need to describe what you did.
  2. How was the control group selected from within the ABIDE dataset? This isn't clear.

Statistical Approach:

I have one major concern with the statistical approach, and that is over-fitting of the multiple regressions. You only have 22 individuals on which to build these models and have one parameter in the smallest model (interpersonal relations) and 5 parameters in the largest models. Given the small n of the study for this, you have far too many parameters for most of the models. 

Additionally, these models are incredibly difficult to read and interpret in a multivariate sense. Your discussion does not really provide any sort of interpretation for the coefficients, but talks about the models more broadly. 

While I am open to your thoughts as to the modeling approach, I would advocate bivariate correlations between the QoL dimensions and the graph theory measures. This could be presented in a correlation matrix, be much easier to read and interpret, and is readily available in R. This would require multiple comparisons correction (recommendation: FDR correction), but I would also advocate reporting corrected and uncorrected p-values.

Bivariate correlations would eliminate the issue of overfitting models and provide future investigators (and you) reasonable targets for future model building with larger samples.

Discussion

The discussion also falls short. While I recognize that there is a paucity of literature on network topology and DS, there are available findings from rs-fMRI. Likewise, while there is limited literature on QoL and rs-fMRI in DS populations, there is research from other populations. The discussion needs to connect this work with previous work (there are only 2 citations in the entire discussion and one is your systematic review), highlight the strengths and identify the weaknesses and provide a reasonable interpretation of the potential meaning of the findings (in light of small n). Overall, the discussion should be updated after the methods are to reflect how you choose to address the concern about multiple regressions.

Author Response

The answers to the reviewer 1 are in the attached document

Reviewer 2 Report

Thanks for the opportunity to review this interesting manuscript. I provide some issues in order to improve this nice manuscript:

-Abstract: According to journal's recommendations, it would be helpful to divide the abstract into background, purpose, methods, results, and conclusions.

-Introduction: Regarding quality of life of Down Syndrome individuals, please consider include information about general quality of life (PLoS One. 2011;6(7):e21879. doi: 10.1371/journal.pone.0021879.), oral health (J Patient Rep Outcomes. 2020 Jun 16;4(1):45. doi: 10.1186/s41687-020-00211-y), and foot health status (Int J Environ Res Public Health. 2018 May 14; 15(5): 983. doi: 10.3390/ijerph15050983), among others, in order to improve this information.

-Methods: The study design section needs to be added. Please, include and cite the followed recommended guidelines and the study type according to the Equatornetwork (https://www.equator-network.org/) recommendations. The completed checklist would be useful as a supplemental file.

-Methods: Please, add a a sample size calculation. This is a mandatory issue that should be addressed.

-Line 176: please, remove "86" as it may be a spelling error.

-Results: Participants sociodemographic data such as age and gender, among others, should be compared between case and control group. This may be a clear limitation to include in the discussion sention

-Discussion: Please, expand your discussion. Two references in your discussion is a very poor indicator. 

Author Response

The answers to the reviewer 2 are in the attached document 

Round 2

Reviewer 2 Report

Thanks for addressing all my comments.